# Nutritional Provision of Iron Complexes by the Major Allergen Alt a 1 to Human Immune Cells Decreases Its Presentation

**DOI:** 10.3390/ijms241511934

**Published:** 2023-07-25

**Authors:** Aila Fakhimahmadi, Ilir Hasanaj, Gerlinde Hofstetter, Clara Pogner, Markus Gorfer, Markus Wiederstein, Nathalie Szepannek, Rodolfo Bianchini, Zdenek Dvorak, Sebastian A. Jensen, Markus Berger, Erika Jensen-Jarolim, Karin Hufnagl, Franziska Roth-Walter

**Affiliations:** 1Comparative Medicine, The Interuniversity Messerli Research Institute, 1210 Vienna, Austria; aila.fakhimahmadi@vetmeduni.ac.at (A.F.); ilir.hasanaj@vetmeduni.ac.at (I.H.); gerlinde.hofstetter@vetmeduni.ac.at (G.H.); nathalie.szepannek@gmail.com (N.S.); rodolfo.bianchini@vetmeduni.ac.at (R.B.); sebastian.jensen@meduniwien.ac.at (S.A.J.); markus.berger@meduniwien.ac.at (M.B.); erika.jensen-jarolim@meduniwien.ac.at (E.J.-J.); karin.hufnagl@meduniwien.ac.at (K.H.); 2Institute of Pathophysiology and Allergy Research, Center of Pathophysiology, Infectiology and Immunology, Medical University of Vienna, 1090 Vienna, Austria; 3Bioresources Unit, Center for Health & Bioresources, AIT Austrian Institute of Technology GmbH, 3430 Tulln, Austria; clara.pogner@ait.ac.at (C.P.); markus.gorfer@ait.ac.at (M.G.); 4Department of Biosciences, University of Salzburg, 5020 Salzburg, Austria; markus.wiederstein@plus.ac.at; 5Department of Cell Biology and Genetics, Faculty of Science, Palacky University, 779 00 Olomouc, Czech Republic; moulin@email.cz

**Keywords:** major allergen Alt a 1, iron, quercetin, nutritional immunity, holo–Alt a 1, enzymatic, immune response

## Abstract

*Alternaria alternata* is a common fungus strongly related with severe allergic asthma, with 80% of affected individuals being sensitized solely to its major allergen Alt a 1. Here, we assessed the function of Alt a 1 as an innate defense protein binding to micronutrients, such as iron–quercetin complexes (FeQ2), and its impact on antigen presentation in vitro. Binding of Alt a 1 to FeQ2 was determined in docking calculations. Recombinant Alt a 1 was generated, and binding ability, as well as secondary and quaternary structure, assessed by UV-VIS, CD, and DLS spectroscopy. Proteolytic functions were determined by casein and gelatine zymography. Uptake of empty apo– or ligand-filled holoAlt a 1 were assessed in human monocytic THP1 cells under the presence of dynamin and clathrin-inhibitors, activation of the Arylhydrocarbon receptor (AhR) using the human reporter cellline AZ-AHR. Human PBMCs were stimulated and assessed for phenotypic changes in monocytes by flow cytometry. Alt a 1 bound strongly to FeQ2 as a tetramer with calculated K_d_ values reaching pico-molar levels and surpassing affinities to quercetin alone by a factor of 5000 for the tetramer. apoAlt a 1 but not holoAlta 1 showed low enzymatic activity against casein as a hexamer and gelatin as a trimer. Uptake of apo– and holo–Alt a 1 occurred partly clathrin-dependent, with apoAlt a 1 decreasing labile iron in THP1 cells and holoAlt a 1 facilitating quercetin-dependent AhR activation. In human PBMCs uptake of holoAlt a 1 but not apoAlt a 1 significantly decreased the surface expression of the costimulatory CD86, but also of HLADR, thereby reducing effective antigen presentation. We show here for the first time that the presence of nutritional iron complexes, such as FeQ2, significantly alters the function of Alt a 1 and dampens the human immune response, thereby supporting the notion that Alt a 1 only becomes immunogenic under nutritional deprivation.

## 1. Introduction

*Alternaria alternata* is a fungus usually residing in plants, where it can either exert endophytic action, meaning it lives in an asymptomatic symbiotic relationship in the plant and contributes in enhancing plants’ health and stress tolerance [1], or it can be an opportunistic pathogen relying on invasion of necrotic tissue [2]. The shift from endophyte to pathogen is thought to occur by abiotic stress (e.g., extreme temperature, nutrient deficiency, radiation, waterlogged stress, salinity) [3] that causes asymptomatic *A. alternata* infection to shift into parasitism, which further aggravates the plant condition [2]. Interestingly, also in humans *Alternaria* can co-exist without causing disease [4] on surfaces, such as the skin [5,6] and the conjunctiva [7], but can elicit disease usually in immunocompromised individuals [6,8,9]. Moreover, it is also strongly associated with human respiratory disease, particularly fungal allergic asthma and rhinitis [10]. Among the *Alternaria* allergens, Alt a 1 is identified as the main sensitizing agent [11,12], suggesting that it may play a driving role in the shift from the apathogenic to pathogenic state in humans as well as plants.

Alt a 1 is a protein shaped as a beta-barrel comprising 11 ß-strands and forms a “butterfly-like” dimer linked by a single disulfide bond with a large dimer interface [13], which has been shown to bind to flavonoid ligands [14]. It can exist as a dimer [13] but easily oligomerize with crystal structures available in PDB repository of Alt a 1 as a tetramer [13] and hexamer [15]. There seems to exist also a pH and ligand dependency [16], in which pH over 6.5, as well as the presence of a ligand, favors oligomerization to tetramers/hexamers. The proposed ligand for Alt a 1 is thought to have a quercetin core [14], which is a flavonoid with known iron-binding capacity [17,18,19,20,21,22], making Alt a 1 very likely a scavenger of iron complexes. Indeed, quercetin can serve as an iron source for commensal and opportunistic microbial pathogens [23], and its anti-oxidative and ROS-scavenging properties are reported to be greater in complex with iron than without iron [24,25,26]. The reported complex stability constant (log ß) for quercetin is 44.2 at physiological pH [27], and thus comparable to the iron affinity of the strongest known microbial catechol-siderophore enterobactin having a complex stability constant of 49 at physiological pH [28]. Moreover, in humans, its impact on human iron homeostasis is well recognized [26,29,30], with therapeutic strategies usually exploiting its function as an iron-chelator [30,31,32,33,34].

We previously suggested that Alt a 1 shares some features with lipocalins [35], such as beta-lactoglobulin [17,20,35,36] and LCN2 [37], as well as the pathogenesis-related protein Bet v 1 [35,36], which are both protein families involved in nutritional immunity and capable of binding to iron when complexed to siderophores or flavonoids.

Consequently, here, we generated recombinant Alt a 1 and assessed its physical and biological attributes. We hypothesized that the immunogenic/pathogenic feature of Alt a 1 is linked to its iron-scavenging ability that it can exert in plants and mammals. We hypothesized that as an innate defense protein involved in nutritional immunity, it would be harmless under nutrient-rich conditions but become immunogenic under nutrient-deprived conditions.

## 2. Results

### 2.1. Alt a 1 Binds with Very High Affinity to Iron–Quercetin Complexes In Silico and In Vitro

The isolated ligand of Alt a 1 has been proposed to contain a quercetin core, with quercetin known to act as a very strong iron chelator [23]. Quercetin alone has anti-inflammatory properties itself, and from previous studies [20,21,36], we hypothesized that the binding of this immunomodulatory molecule in conjunction with iron would hint towards a function of Alt a 1 in nutritional immunity. Binding of this ligand transforms Alt a 1 from apo– (empty) to holo– (filled) Alt a 1. In the first step, we proved by different means that Alt a 1 binds to iron quercetin complexes (FeQ2, FeQ3). In contrast to other allergenic molecules, the calyx of Alt a 1 is too small for incorporation of FeQ2 within one protein. However, binding has been suggested at the dimeric interface and thus may affect the quaternary state of Alt a 1. In silico analysis revealed that Alt a 1 may be indeed capable of binding very strongly to FeQ2, and that the affinity towards FeQ2 increases from monomer to tetramer/hexamer, resulting in outstanding theoretical affinities to FeQ2 (and FeQ3). The calculated affinities to FeQ2/FeQ3 reached even the picomolar range and thus can be considered as strong as high-affinity antibodies to their cognate antigen, with their binding capacity usually ranging in the pico to femtomolar range (Figure 1). Calculated FeQ2 and FeQ3 binding affinity energy to the tetramer was −13.526 and −13.966 kcal/mol, respectively, corresponding to a dissociation constant K_d_ for both of about 100 pM, whereas binding of the tetramer to quercetin alone had “only” a calculated affinity energy of −8.555 kcal/mol, corresponding to a K_d_ of 530 nM. Similarly, also the hexamer bound with an outstanding calculated affinity of −11.227 to FeQ2 and −12.224 to FeQ3, corresponding to a K_d_ of 6 and 1 nM, respectively, whereas solely quercetin had a binding affinity of about −7.438 kcal/mol and a K_d_ of 3.5 μM.

Moreover, by spectroscopical means, iron–quercetin complex (FeQ2) formation was visible (Figure 2) and could be measured spectroscopically, with a prominent second maximum appearing at about 450 nm appearing upon complex formation (light blue line). Indeed, quercetin has per se a light-yellow colour, which results in a spectrum peaking with a single maximum at about 340 nm at physiological pH. In contrast, when in complex with iron (at a pH > 6.5), the colour turns brownish, and due to the interaction of the catechol-group with the d-orbitals of ferric iron, a second maximum, usually about 450 nm, appears [17,21,24]. Binding to FeQ2 could be proven because this second absorption maximum diminished concentration dependently upon the addition of Alt a 1 (empty protein, blue lines) (Figure 2a). Alt a 1 was also able to bind to quercetin alone (first maximum about 330 nm, grey lines). However, here, a concentration dependency was not seen. CD analysis revealed that upon ligand binding, no shift in the minima of Alt a 1 occurred, indicating that the secondary structure was only slightly affected by FeQ2 (Figure 2b).

Alt a 1 is usually dimeric but also forms tetra- and hexamers, with the quaternary state being reported to be pH-dependent (monomers/dimers below pH 6, otherwise tetra- and hexamers) [16]. Our in silico analysis revealed particularly strong binding of Alt a 1 as a tetramer to FeQ2. As such, we performed dynamic light scattering analysis to assess whether pH or the ligand FeQ2 affected oligomerization. Here, % intensity was used as an approximation for the size distribution in the solution [38]. With and without a ligand, the quaternary state was similar between pH 5.5 and 7.2 in apo– and holoAlt a 1 (Figure 1c). However, when comparing % intensity of apoAlt a 1 and holoAlt a 1 at pH 6.5 or 7.2 on a linear scale, a greater degree towards oligomerization was apparent (Figure 2d). To sum up, we provide evidence that Alt a 1 binds particularly strongly to iron–quercetin complexes as a tetramer in a physiologically relevant pH.

### 2.2. Alt a 1: Enzymatic Activity, Transport Function, and Clathrin-Dependent Uptake

*Alternaria* is considered an opportunistic fungal pathogen that lives on plants without causing any illness until “circumstances “dictate their host invasion ability and their pathogenicity. Having established its binding to iron complexes, we thus next addressed the potential function of Alt a 1 in nutritional immunity as this would imply different functional features depending on whether Alt a 1 is under nutrient-poor or nutrient-rich conditions.

Most innate proteins harbor several functions and often possess enzymatic activity. Thus, we assessed first whether Alt a 1 possessed any enzymatic and metalloproteolytic activities by casein and gelatin zymography. Silver staining of uncooked recombinant Alt a 1 in gels run under non-reducing conditions revealed that Alt a 1 is predominantly present as oligomer (tetra-/hexamer) at neutral pH and without heat treatment, which seemed unaffected by the presence of ligands, such as iron–quercetin or zinc and calcium. In contrast, upon cooking, disruption of the hexameric state is visible, resulting in a predominant band representing dimers at about 25 kDa (Figure 3a). We indeed describe here that as a hexamer, Alt a 1 without ligands (apoAlt a 1) has a weak concentration-dependent enzymatic activity and is capable of hydrolyzing peptide bonds at around 100 kDa, however, only in its apo−form and not when it already binds to iron–quercetin (holoAlt a 1) (Figure 3b,c). Similarly, we report that also a trimeric form possesses a weak concentration-dependent collagenase activity in the presence of zinc and calcium and thus can act as calcium-dependent zinc-containing endopeptidase/metalloproteinase, but again, only in nutrient-deprived conditions as apoAlt a 1 and not as holoAlt a 1 (Figure 3b,c). As such, we show here that the potential pathogenicity of Alt a 1 is increased under nutrient-deprived conditions and depends on its quaternary structure.

In the next step, we assessed the ability of Alt a 1 to impact human immune cells depending on whether it carried iron–quercetin complexes or not. This was assessed by measuring iron levels in human monocytic THP1 cells. Here, the quenching of the calcein signal serves as a measure of the cytosolic iron [39]. Our analysis revealed that the cytosolic labile iron content in THP1 cells was significantly decreased upon apoAlta 1 exposure (resulting in a higher intensity of the calcein signal), indicating intracellular iron-scavenging abilities by apoAlt a 1, whereas this was not the case when incubation occurred with holoAlt a 1 (Figure 3d). Moreover, the action of the quercetin-ligand, as a known activator of AhR, can be measured to assess uptake of iron–quercetin complexes upon incubation with apo−/holo−Alt a 1 using the human AZ-AhR cellline [40] that is stably transfected with several aryl hydrocarbon-receptor AhR binding sites upstream of the luciferase reporter gene [40]. Here, we demonstrate that Alt a 1 facilitated quercetin-transport into human cells and that the AhR activity concentration dependently increased by the addition of Alt a 1 from 1.0 ± 0.3 with FeQ2 alone, to 2.7 ± 0.8 with 0.5 μM, to 5.26 ± 1.31 with 1 μM Alt and to 2.1 ± 0.3 with 1.5 μM Alt a 1 (Figure 3e). We thus conclude that while apoAlt a 1 further deprived monocytes of iron, holoAlt a 1 transported these beneficial ligands into human cells.

We further assessed mechanistic uptake pathways into human monocytic cells with at least six pathways of endocytosis being described, ranging from (a) clathrin-coated pits, which are clathrin and dynamin-dependent and usually indicative of being receptor-mediated [41], to (b) fast endophilin-mediated endocytosis, which is clathrin-independent but dynamin-dependent for rapid ligand-driven endocytosis of specific membrane proteins, to (c) caveolin endocytosis, to (d) clathrin-independent carrier, (e) micropinocytosis and (f) phagocytosis [41]. As depicted in Figure 3f,g, in our experiments with THP1 cells, increased binding of fluorescent labeled Alt a 1 was observed at 38 °C compared to 4 °C, suggesting internalization of Alt a 1. There was a trend (*p* = 0.0736) that uptake for apoAlt a 1 was better, compared to holoAlt a 1. When blocking clathrin-mediated uptake with Pitstop2 or dynamin-mediated incorporation with Dyngo4, we saw, irrespective of whether cells were incubated with apo− (Figure 3h) or holo−Alt a 1 (Figure 3i), a decrease—but not complete blocking—in Alt a 1 uptake.

As such, we conclude that Alt a 1’s cargo and quaternary state, as well as the environmental/host conditions, impact the function of Alt a 1: The empty, nutrient-deprived version has an enzymatic function, decreases the cytosolic iron content and cellular uptake occurrs partly via clathrin-coated pits and likely via specific receptors. In contrast, Alt a 1 with iron–quercetin complexes impeded Alt a 1’s intrinsic enzymatic activity. Here, uptake also seems to occur partly via clathrin-coated pits, with the cargo being transported into monocytic cells resulting in activation of the AhR pathway.

### 2.3. Decreased Antigen Presentation upon Incubation with Alt a 1 Binding to Iron–Quercetin Complexes

In the next step, we incubated apo− or holoAlt a 1 with peripheral blood mononuclear cells (PBMCs) of *Alternaria* allergic donors to assess their direct impact on primary human immune cells. As iron–quercetin was used as ligand, cells were only incubated overnight in iron-free and serum-free media to be able to discern between the impact of Alt a 1 without or with ligands. Indeed, overnight incubation with Alt a 1 resulted in a significant decrease of CD14+ surface expression, which was particularly prominent when Alt a 1 was in conjunction with iron–quercetin complexes (holoAlt a 1) (Figure 4a,b). This was accompanied by significantly fewer CD14+ cells expressing the costimulatory molecules CD86 (Figure 4c) as well as HLADR (Figure 4d) and a decreased frequency of CD14+cells expressing both HLADR+CD86+ (Figure 4e). As such, we show that ligand shuttling of holoAlt a 1 particularly decreased the frequency of competent antigen-presenting cells expressing costimulatory molecules, which was not due to increased cell death (Appendix A).

We also assessed cytokines released into the media after 18 h, which are derived due to the short incubation time span predominantly from monocytes or B cells. As no further stimuli were used, only very low levels of cytokines were detected at all, with the highest levels being for the inflammatory and macrophage-associated cytokines IL6, which was significantly elevated upon incubation for both Alt a 1 forms. Further, significantly elevated levels of TNFα, but also IL10 cytokines compared to medium, were detected in PBMCs exposed to holoAlt a 1, whereas apoAlt a 1 stimulation was associated with increased IL5 and IL13 secretion compared to unstimulated cells (Figure 5).

Thus, in vitro, the immune response considerably differed upon Alt a 1 exposure, whether it bound iron complexes or not, with the iron-loaded form considerably reducing the frequency of competent antigen-presenting cells and leading to an enhanced IL10 production. In contrast, Alt a 1 without any ligand rather promoted a Th2-associated milieu with elevated IL5.

## 3. Discussion

*Alternaria* is usually a plant fungus that emerges under certain conditions as a pathogen. It is not considered very thermotolerant [42]. Still, the presence of *Alternaria* has been detected in healthy individuals in the oral mucosa [43,44]. However, also pathogenic cutaneous, subcutaneous, as well as mucosal infestations affecting even the lungs [42] have been reported, particularly in immunocompromised individuals [6,45,46,47]. Moreover, its contribution to human asthma [48] and, to a lesser extent also to atopic dermatitis, [49,50] is well recognized. Here, the major associated immunogenic protein identified with asthma is the major allergen “Alt a 1” to which most affected individuals generate IgE and IgG antibodies [49].

The association of Alt a 1 with disease and our immune system as well as its ability to bind to flavonoids, which are potent iron carriers [27,51,52], suggest that it may be involved in nutritional immunity. In particular, Alt a 1 shares its flavonoid-binding properties with other allergens from birch with Bet v 1 [17,53], hazelnut with Cor a 1 [54], strawberry with Fra a 1/Fra a 3 [55], peanut with Ara h 2/Ara h 6 [56], Ara h 8 [57,58], and Ara h 1 [59] and with cow Bos d 5 [21,22], all of which are known as innate defense proteins in the respective plant/mammals. For many of these allergens, also their iron-binding features have been described [17,21,22,36], and thus, not surprisingly, we report here up to picomolar theoretical affinities of Alt a 1 to iron–quercetin complexes. Indeed, the affinity of Alt a 1 to the iron complex FeQ2 surpasses the ones already reported for other allergens [17,18,21,60] by at least a factor of 10 and joins the rank with the affinity reached by high-affinity antibodies after affinity maturation [61,62,63,64,65,66]. Although the affinities have only been calculated by in silico means and only give an approximation of the “real” affinity, the calculating software AutoDock Vina [67,68] has been validated in numerous studies. Consequently, it is used nowadays as a tool for screening potential protein–ligand interactions and predicting their affinities [69,70,71,72,73,74,75].

In line with the in silico data, we show that Alt a 1 binds very effective iron complexes and that ligand binding seems to promote oligomerization. Indeed, Alt a 1-like proteins have already been reported to induce plant defense responses and are suggested to be involved in nutrient acquisition by inducing necrosis [76]. Moreover, uptake of apoAlt a 1 was capable of reducing cytosolic iron further, highlighting its ability as an iron scavenger.

Interestingly, we show here that trimeric and hexameric Alt a 1 seems to harbor endopeptidase and metalloproteinase activity, but only when in a nutrient-deprived apo−state. We do not attribute, but cannot completely exclude that the displayed enzymatic activities of the recombinant Alt a 1 is due to remaining impurities in the same molecular size and with similar physico-chemical properties as Alt a 1, because also other groups have reported that Alt a 1 displays enzymatic phosphatase, phosphoamidase and esterase activity [77] and also our CD analysis gave us no indication of “other” proteins. We report here, for the first time, that the enzymatic features of Alt a 1 seems to depend on the quaternary state as neither the dimer nor the tetramer showed any enzymatic activity, but the enzymatic activity was restricted to the trimer/hexamer. The enzymatic activity of Alt a 1 was rather low, suggesting, nonetheless, that only under certain conditions (when Alt a 1 does not bind any ligand and trimer and/or hexamerization occurs), enzymatic functions may be promoted. This may be in situations in which the nutrient flow is disrupted, such as when the fruit has fallen from the tree, or in stressful situations of the plant/human that further restrict access to nutrients.

In our study, Alt a 1 was also differently handled by human immune cells as uptake of the apo− and nutrient-deprived version tended to be more efficient than the holo−version. Uptake was clathrin- and dynamin-dependent in both versions, implying receptor-specific uptake, which is supported by reports detailing uptake of Alt a 1 via the LCN2-receptor SLC22A17 [78] and also via TLR2 and TLR4 [79] in the airways. As such, though considered foreign in humans, Alt a 1 seems to enter the body by specific receptors.

Interestingly enough, when assessing its impact on primary immune cells, we particularly saw an immunosuppressive monocytic phenotype when uptake occurred of Alt a 1 carrying iron complexes. This was followed by a significant suppression, not only of CD14 but also of costimulatory proteins such as CD86, which indicates decreased antigen presentation capacity. The reason for this pronounced reduction in its ability to present antigens might be related to the active transport of quercetin by holoAlt a 1. Indeed, activation of the cytoplasmic promiscuous AhR by quercetin [80] is described to mediate anti-inflammatory stimuli [81] and keeps antigen-presenting cells, such as dendritic cells or macrophages, in an immature state [82].

Under nutrient-deprived conditions, both forms, apo and holoAlt a 1, were immunogenic and led to IL6 secretion. However, only holoAlt a 1 carrying iron complexes induced TNFα and IL10, whereas apoAlt a 1 incubation was rather associated with a Th2 signature and elevated IL5 [83]. The iron scavenging features of apoAlt a 1 were associated with a Th2-milieu, which is in accordance with the literature showing that iron deficiency is associated in vitro [84,85,86] and in human clinical studies with Th2 inflammation [20,87,88,89,90,91,92,93]. To stress out the importance of iron, reduced immunogenicity has also been shown upon iron-binding with Arah1, Arah3/Arah4 allergens [94], ovotransferrin [95], beta-lactoglobuline [21], and Bet v 1 [17]. In this context, we report with Alt a 1 another major allergen that exhibits these iron-binding features, which appear to determine whether Alt a 1 is pathogenic or tolerogenic.

To summarize, we reveal, for the first time, that iron quercetin binding reduces the immunogenicity of Alt a 1. We provide evidence that both iron and quercetin are shuttled by Alt a 1 into human immune cells, thereby hindering activation of monocytes. The present data are in line with our findings that Alt a 1 can transport micronutrients and that only the nutrient-deprived protein is immunogenic.

## 4. Materials and Methods

### 4.1. Materials

Quercetin, iron, and deferoxamine mesylate were purchased from Sigma (Sigma Aldrich, Steinheim, Germany). Ficoll-Paque PLUS was from GE Healthcare (Uppsala, Sweden) and Legendplex Th1/Th2 Assay Kit was from from BioLegend (San Diego, CA, USA).

### 4.2. In Silico Structural and Docking Analysis

Atom coordinates of Alt a 1 were taken from the high-resolution 1.9 Å crystal structure with Protein Data Bank (PDB) code 3V0R [13]. Atoms of water, ligands, cofactors, and ions were removed. The geometries of quercetin were obtained upon energy minimization with the MM2 force field of initial structures drawn using the ChemBioDraw/ChemBio3D Ultra 12.0 package. Docking input files for protein and ligands were prepared with reduced v3.23 [96], the ADFR software suite (build 5, 28 October 2019; https://ccsb.scripps.edu/adfr (accessed on 12 August 2021), and AutoDockTools [97]. Docking calculations were performed with AutoDock Vina v1.2.3-mod [67,68]. The docking solution with the lowest affinity energy E_aff_ was selected. Estimates of dissociation equilibrium constants K_d_ were then calculated for the protein–ligand complexes by assuming E_aff_ ~ ∆G with K_d_ = exp(−∆G/RT) at T = 298.15. Protein structural visualizations were prepared with UCSF Chimera [98]. Close-up views of the ligand binding site were defined by a 4 Å radius around the FeQ2 complex in the structure of Alt a 1, with hydrogen bonds shown as broken lines.

### 4.3. Alt a 1 Expression in Pichia Pastoris

*E. coli* DH5α strain was grown in low-salt Luria–Bertani broth and transfected with the pPICZα A vector (Invitrogen, Waltham, MA, USA) containing the synthetic Alt a 1 gene (General Biosystems, Morrisville, NC, USA) via cloning with *EcoRI* and *NotI*, then selected with Zeocin™ antibiotic (Invitrogen, Waltham, MA, USA). The plasmid DNA was isolated and checked with PCR using AOX1 gene forward and reverse primers: 5′ AOX1 5′-GACTGGTTCCAATTGACAAGC-3′ and 3′ AOX1 5′-GCAAATGGCATTCTGACATCC-3′. Next, the plasmid DNA was linearized with *SacI* and transferred to the Pichia pastoris X33 strain using electroporation and cultured on Yeast Extract Peptone Dextrose Medium (YPD) and +Zeocin™ plates. Integration of the expression construct into the genome of the Pichia transformants was confirmed by (1) PCR analysis of colonies using plasmid- and gene-specific primers as well as by (2) DNA sequencing. A Mut positive colony was selected and inoculated on BMGY (1% yeast extract, 2% peptone, 100 mM potassium phosphate, pH 6.0, 1.34% YNB, 4 × 10^−5^% biotin, and 2% glycerol) and Zeocin™ (25 μg/mL). They were cultured at 27–28 °C in a shaking incubator (250–300 rpm) until the culture reached an OD600 of 2–6. The cells were then harvested. To induce expression, the cell pellet was resuspended in BMMY medium (the same as BMGY with 1% methanol instead of glycerol) using 1/5 to 1/10 of the original culture volume and 100% methanol was added to a final concentration of 1.5% every 24 h to maintain induction. Final harvesting was done after 144 h. The supernatant was collected and purified with anion exchange chromatography using 10 mM Na-phosphate buffer (pH 7.5). SDS-PAGE and circular dichroism (CD) spectroscopy were used to verify protein purity, identity, and secondary structure. Measurement of endotoxin content was done by EndoZyme recombinant Factor C Endotoxin Detection Assay (Biomerieux, Craponne, France) and total protein content by BCA assay (Pierce BCA Protein Assay Kit, Thermo Scientific, Rockford, IL, USA) according to the manufacturer’s instructions [99,100]. Batches were sterile-filtered (0.22 μM) before use. In this manuscript, two different batches of recombinant Alt a 1 were used. PBMC analysis was conducted with Alt a 1 batch 1 (concentration of stock: 1.1 mg/mL containing 2.07 EU/mL endotoxin = 1.9 EU/mg protein), and zymography was conducted with batch 3 (concentration of stock: 3.1 mg/mL, 54.9 EU/mL = 17.7 EU/mg protein).

### 4.4. Generation of apo– and holoAlt a 1

Alt a 1 (1–3 mg/mL) was dialyzed three times against 10 μM deferoxamine mesylate salt (Sigma D9533), following dialyzation against deionized water to generate apoAlt a 1. HoloAlt a 1 was generated by adding pre-formed iron–quercetin complexes (FeQ2). Three mM FeQ2 complexes were generated by dissolving quercetin (Sigma 1592409) in 1 M NaOH and adding acidic iron (Iron-standard AAS, Sigma 16596) at a ratio 2:1, before adding them to apoAlt a 1 to a final concentration of, e.g., 2.5 μM Alt a 1, 5 μM quercetin and 2.5 μM iron or dilutions thereof.

### 4.5. Spectral Analysis of Alt a 1 with Iron–Quercetin Complexes

For the spectral analyses, a physiological saline solution (0.89% NaCl) was used as a buffer to minimize iron contamination from the air. The pH was adjusted to a pH of about 7 for Figure 2a,b. Optical density was measured using (1) 80 μM quercetin (=40 μM Q2) alone or 40 μM FeQ2 with increasing concentrations of Alt a 1 (2/4/8 μM) with a UV-VIS spectrometer (Tecan InfiniteM200 PRO, Tecan, Männedorf, Switzerland). All measurements were repeated at least three times with similar results.

Circular dichroism (CD) measurements were performed on a Jasco J-715 spectropolarimeter with 100 μg/mL sample using a 1-mm path-length quartz cuvette equilibrated at 20 °C as already described [101]. Spectra were recorded from 190 to 260 nm with 0.2 nm resolution at a scan speed of 50 nm/min and resulted from the average of five scans. The final spectra were corrected by subtracting the corresponding baseline spectrum obtained under identical conditions. Results were expressed as the mean residue ellipticity (Θ) at a given wavelength.

Dynamic light scattering (DLS) was employed to assess the quaternary structure of the apo/holoAlt a 1 protein with FeQ2 at pH 4, 5.5, 6.5, 7.2, and 8 using 10 mM acetate/150 mM NaCl buffering system for pH 4 and pH 5.5 and 10 mM Hepes-NaOH buffer/150 mM NaCl for pH 6.5, pH 7.2, and pH 8. 5 μM apoAlt a 1 was incubated with 8 μM FeQ2 for 5 min before removing unbound ligand via spin columns (pore size 0.22 μm, Corning-Costar-Spin-X, CLS8160, Corning, NY, USA) and centrifugation for 5 min at 1000× *g*. Subsequently, samples were transferred to cuvettes (UVette^®^ 220–1.600 nm, Eppendorf, Hamburg, Germany), and DLS measured using the Dynapro Nanostar DLS instrument (Wyatt Technology, Santa Barbara, MI, USA).

### 4.6. Silverstain, Casein, and Gelatine-Zymography with apo–/holoAlt a 1

Ten μL recombinant apo– and holoAlt a 1 in 10 mM Hepes-buffer/150 mM NaCl pH 6.5 (1 mg/mL) with or without 2 μL 10× Zn/Ca Tris Buffer (500 mM Tris, 10 mM ZnCl2, 50 mM Ca) were mixed with 5 μL non-reducing 4× sample buffer pH 6.8. Samples were left uncooked or were cooked for 5 min at 95 °C before they were separated on Mini-Protean TGXStain-Free Gel 4–20% (Bio-Rad 4568095, Biorad, Hercules, CA, USA) electrophoresis. Gels were subsequently fixed, and silver stained upon addition of thiosulfate solution (0.02%), silver nitrate reagent (0.02% silver nitrate, 0.02% formaldehyde), developing in 3% sodium carbonate containing 5 mg/L thiosulfate and stopping development with 0.5% glycine solution.

Zymography was performed as previously described [102]. Briefly, 0.5, 1, 2, and 3 μg apo–/holoAlt a 1 were mixed with 2× sample buffer (20% glycerol, 4% sodium dodecyl sulfate: SDS, 125 mM Tris-HCl, 0.01% bromophenol blue, pH 6.8) with neither 2-mercaptoethanol nor boiling. Samples were separated at 4 °C in 15% SDS-polyacrylamide gel electrophoresis (SDS-PAGE) copolymerized with 0.1% casein or gelatin. Migrated enzymes were renatured from SDS by soaking in 2.5% Triton X-100 for 30 min twice and then activated in 50 mM Tris-HCl, pH 8.0, for casein, 50 mM Tris-HCl, 5 mM CaCl2, 1 mM ZnCl2, pH 7.6, for gelatin, at 37 °C for 16 h. Gels were stained with 0.2% Coomassie Brilliant Blue (CBB) R-250 solution. Digestion bands were viewed with ChemiDoc Touch Imaging system (Biorad, Hercules, CA, USA) and analyzed using the Java-based image processing program ImageJ (The National Institutes of Health, Bethesda, MD, USA).

### 4.7. Endocytosis Experiments with Thp-1 Cells and Labile Iron Measurements

An Alexa Fluor 647-tag was added to apo/holoAlt a 1 according to the manufacturer’s protocol (Component B, Zip Alexa Fluor 647 Rapid Antibody Labeling Kit, Invitrogen, Waltham, MA, USA) by adding Component B to the sample. Cells were harvested and washed with 0.89% NaCl and then resuspended in DMEM without phenol red, L-Glutamine, and FCS to a concentration of 1 × 10^6^ cells/mL, of which 200 μL each were transferred to flow cytometry tubes. After 2 h of starvation time, in designated wells, 5 μM Pitstop-2 were added for 30 min or 20 μM Hydroxy-Dynasore for 15 min or 0.4% DMSO for 30 min as control, prior addition of 1 μM APC-labeled apo/holoAlt a 1 or medium for 1 h. Subsequently, cells were stained with 5 μM CalceinAM-Violet for 15 min and analyzed by flow cytometry. Doublets were excluded before measuring the mean fluorescence intensity (MFI) of APC and calcein violet. Acquisition and analyses were performed on a FACS Canto II machine (BD Biosciences, San Jose, CA, USA) using the FACSDiva Software 6.0 (BD Biosciences).

### 4.8. AhR Reporter Assay

AZ-AhR assay was done as previously described [18]. Briefly, AZ-AhR cells were plated on 96-well plates at a density of 2 × 10^4^ cells/well for 18 h. Subsequently, cells were stimulated for 18 h in triplicates with 10 μM of iron–quercetin complexes alone or in addition with 0.5, 1, or 1.5 μM Alt a 1. Cells were washed once with 0.89% NaCl before lysis buffer of the luciferase assay kit (Promega E4530,Promega, Madison, WI, USA) was added. After a single freeze-thaw cycle, 20 μL/well of lysates were transferred into a black 96-well flat-bottom plate (Thermo Scientific), and bioluminescent reaction was started with addition of 100 μL/well of luciferase assay reagent (Promega). Chemiluminescence was measured (10 s/well) using a spectrophotometer (Tecan InfiniteM200 PRO). Experiments were repeated thrice, with data on two independent experiments shown as normalized to medium levels.

### 4.9. Flow Cytometric Analysis of Human PBMCs

PBMCs were obtained from *Alternaria*-allergic patients after written informed consent. PBMCs were isolated by Ficoll–Paque (GE Healthcare) and washed with 0.9% NaCl before incubation with 5 μM iron quercetin (FeQ2) alone and/or in combination with 2.5 μM apoAlt a 1 in media containing neither phenol red nor FCS for 18 h as previously described [21]. Subsequently, cells were stained with combinations of CD14-APC-Cy7 (Biolegend, clone M5EZ), HLADR-PE (Biolegend, San Diego, CA, USA, clone L243PC), and CD86-PE-CY7 (Biolegend, clone IT2.2) for flow cytometric analysis. Doublets were excluded before gating on CD14+ cells in the monocytic population, and consecutive gating for HLADR+, CD86+, or HLADR + CD86+. Acquisition and analyses were performed on a FACS Canto II machine (BD Biosciences, San Jose, CA, USA) using the FACSDiva Software 6.0 (BD Biosciences).

### 4.10. Cytokine Analysis of PBMCs Supernatant

Cytokines of PBMC supernatant were analyzed via Legendplex Th1/Th2 Assay Kits (BioLegend) according to the manufacturer’s protocols and having a reported sensitivity of 2 to 4 pg/mL.

### 4.11. Statistical Analysis

Statistical analysis of zymography data was carried out by RM one-way ANOVA followed by Tukey’s multiple comparison test with a single pooled variance. Cell culture experiments with THP1 cells were analyzed by mixed-effect analysis with the Geisser–Greenhouse correction, followed by Tukey’s multiple comparison test. Concentration dependency and the differences in treatments were analyzed with RM two-way ANOVA with the Geisser–Greenhouse correction and Tukey’s multiple comparison test using a single pooled variance. The stimulated peripheral blood mononuclear cells were analysed by RM one-way ANOVA, with the Geisser–Green house correction and Tukey’s multiple comparison test, with individual variances computed for each comparison. For the cytokines statistical analysis, matched data were analyzed with non-parametric Friedman test, followed by Dunn’s multiple comparison test. A *p*-value below <0.05 was considered significant.

## 5. Patents

F.R.-W. and E.J.-J. declare inventorship of EP2894478 (Method and means for diagnosing and treating allergy by Roth-Walter F, Gomez-Casado C, Jensen-Jarolim E, Pacios LF, Diaz-Perales A, Singer J”. EP 14150965.3, Year: 01/2014; US 14/204,570) (owned by Biomedical International R+D GmbH, Vienna, Austria). E.J.-J. declares shareholdership in Biomedical Int. R + D GmbH, Vienna, Austria.

## Figures and Tables

**Figure 1 ijms-24-11934-f001:**
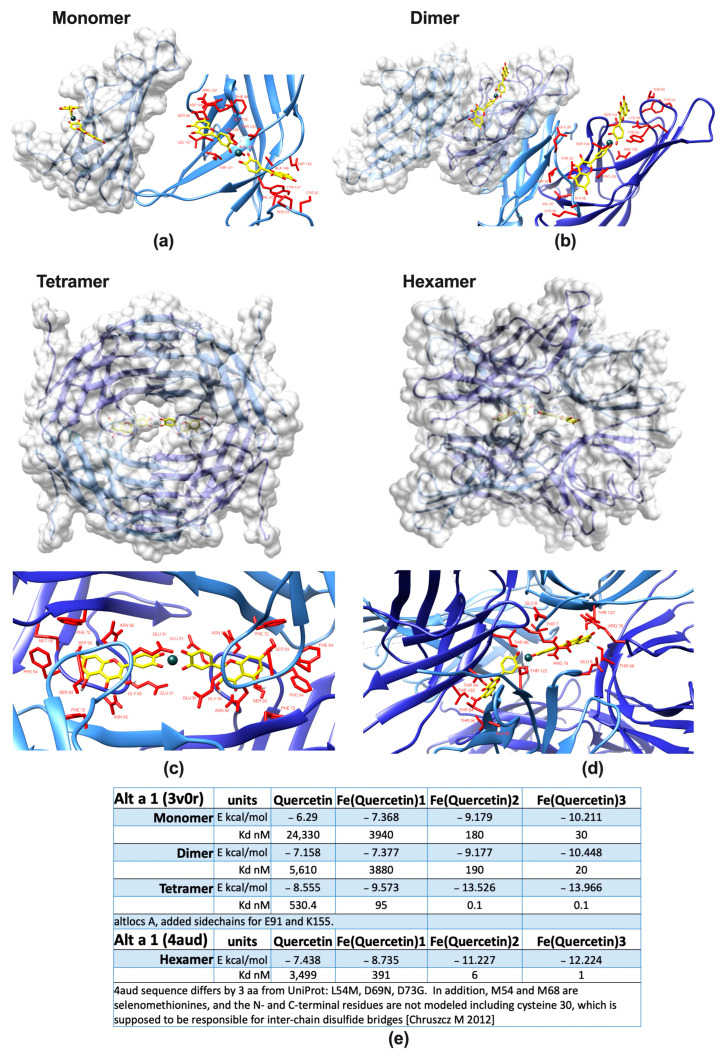
Alt a 1 binds quercetin stronger in conjunction with iron. Protein surface of the major *Alternaria* allergen Alt a 1 and a close-up of amino acids (red) in 4 Å vicinity of iron(quercetin)_2_ (FeQ2) complexes as a (**a**) monomer, (**b**) dimer, (**c**) tetramer, (**d**) hexamer [13] with FeQ2 (sticks with carbons in yellow, oxygens in red, and iron shown as a grey sphere). (**e**) Summary of calculated affinities as derived from in silico docking of quercetin in conjunction with iron to mono-, di-, tetra-, and hexameric Alt a 1.

**Figure 2 ijms-24-11934-f002:**
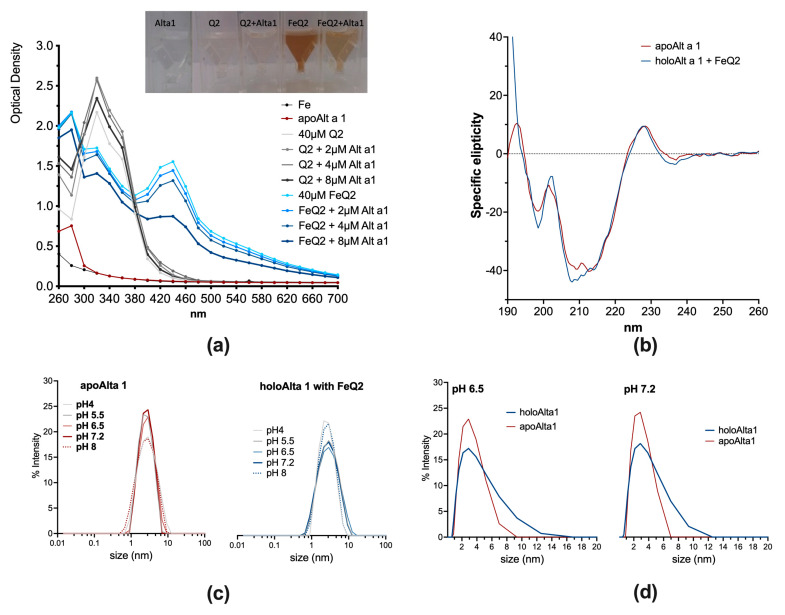
Ligand binding hardly affects the secondary structure of Alt a 1 but may promote oligomerization. (**a**) Optical spectra of 100 μM Q2 or FeQ2 alone, or with increasing concentrations of Alt a 1. (**b**) Circular dichroism analysis of apo− and holoAlt a 1. (**c**) Dynamic light scattering of apo− and holo−Alt a 1 at pH 4, 5.5, 6.5, 7.2, and 8. (**d**) Comparison of oligomerization of apo− and holoAlt a 1 at pH 6.5 and 7.2.

**Figure 3 ijms-24-11934-f003:**
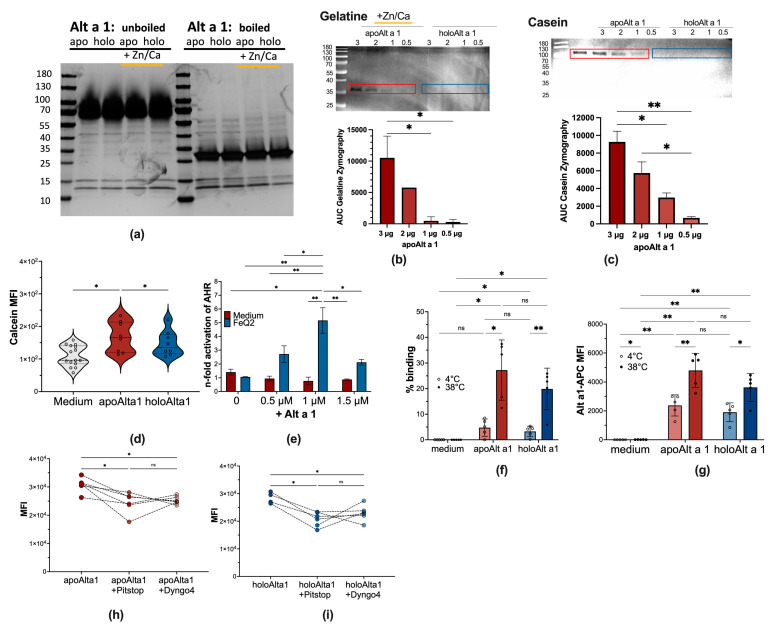
apoAlt a 1 exerts proteolytic function and decreases the labile iron pool upon clathrin-mediated uptake. (**a**) Recombinant Alt a 1 (10 μg) in the absence (apo, red box) or presence of iron–quercetin (holo, blue box) and with or without Zn/Ca were left uncooked or cooked before proteins were separated on 4–20% SDS-PAGE under non-reducing conditions and proteins bands were visualized by silver stain. (**b**) Representative inverted image of casein zymogram with summary of quantification of 2 independent experiments. (**c**) Representative inverted image of gelatin zymogram with summary of quantification of 2 independent experiments. (**d**) Summary of mean fluorescence intensity MFI of the calcein signal in THP1 cells incubated 18 h with apo− and holoAlt a 1 from three independent experiments (**e**) AhR activation was measured after 18 h in AZ-AhR cells treated with 10 μM of iron–quercetin complexes alone or in combination with 0.5, 1, and 1.5 μM Alt a 1. Experiments were performed at least 3 times independently and normalized to medium alone. Summary of 2 h-starved THP1 cells incubated for 1 h at 4 °C or 38 °C with fluorescent-labeled apo/holoAlt a 1 and determined by flow cytometry (**f**) % binding (**g**) APC-mean fluorescence intensity of Thp1-cells, (**h**) incubated for 1 h with labeled apoAlt a 1 in the presence of Pitstop 2 or Dyngo4 and (**i**) for 1 h with labeled holoAlt a 1 in the presence of Pitstop 2 or Dyngo4. Statistical analyses of zymograms in (**b**,**c**) from two independent experiments were analyzed by RM one-way-ANOVA followed by Tukey’s multiple comparison test with a single pooled variance, (**d**,**h**,**i**) was analyzed by Mixed-effect analysis with the Geisser–Greenhouse correction, followed by Tukey’s multiple comparison test, (**e**–**g**) was analyzed with RM two-way Anova with the Geisser–Greenhouse correction and the Tukey’s multiple comparison tests with a single pooled variance. * *p* < 0.05, ** *p* < 0.01.

**Figure 4 ijms-24-11934-f004:**
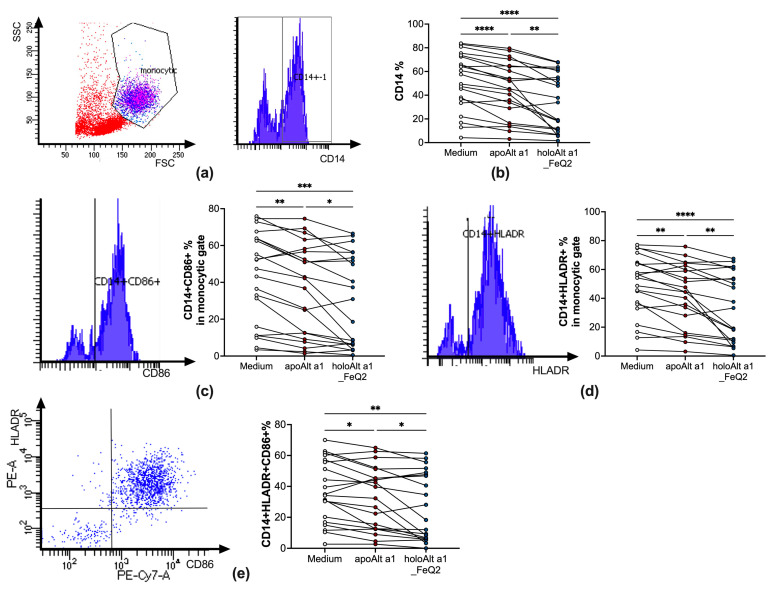
HoloAlt a 1 significantly reduces the frequency of competent antigen-presenting cells. Flowcytometric analyses of PBMCs (*n* = 20) stimulated with medium (white circles), apoAlt a 1 alone (red circles) or in combination with iron–quercetin (holoAlt a 1_FeQ2, green circles)). (**a**) Gating strategy and (**b**) relative numbers of CD14+ in the monocytic gate, with further gating to (**c**) CD86+ or (**d**) HLADR+ with the respected summary of relative numbers in the monocytic gate. (**e**) Gating of CD14+ cells for HLADR + CD86+ expression and summary of relative numbers thereof. RM one-way ANOVA, with the Geisser–Green house correction, followed by Tukey’s multiple comparison test, with individual variances computed for each comparison. * *p* < 0.05; ** *p* < 0.01,*** *p* < 0.001; **** *p* < 0.0001.

**Figure 5 ijms-24-11934-f005:**
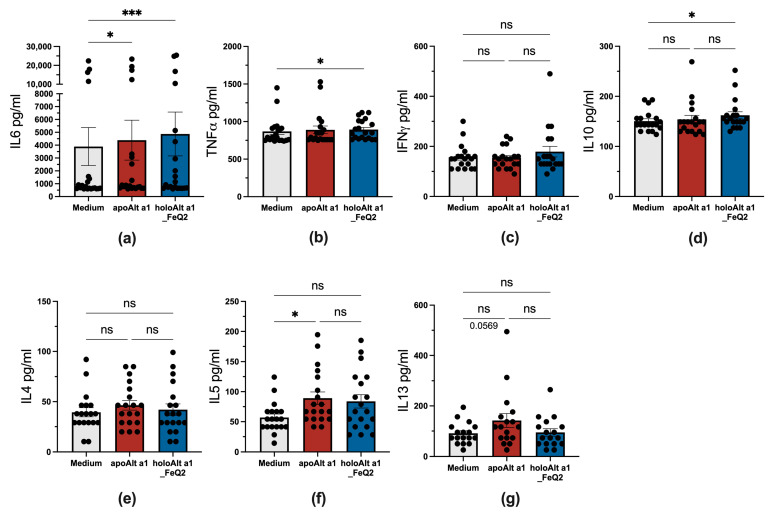
Distinct cytokine profile upon apo– and holoAlt a 1 stimulation of PBMCs from *Alternaria* allergic donors. Supernatant of PBMCs (*n* = 20) stimulated in the presence or absence of Alt a 1 with or without FeQ2 and assessed for (**a**) IL6, (**b**) TNFα, (**c**) IFNγ, (**d**) IL10, (**e**) IL4, (**f**) IL5, and (**g**) IL13. For statistical analysis, matched data were analyzed with a non-parametric Friedman test, followed by Dunn’s multiple comparison test. * *p* < 0.05; *** *p* < 0.001, n.s., not significant.

## Data Availability

The data presented in this study are all contained within the article.

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
