# Peer review of "Nutritional Provision of Iron Complexes by the Major Allergen Alt a 1 to Human Immune Cells Decreases Its Presentation"

_ijms, 2023, doi:10.3390/ijms241511934_

Round 1

Reviewer 1 Report

Review report: Nutritional provision of iron complexes by the major allergen Alt a 1 to human immune cells decreases its presentation

The authors investigated whether binding of iron quercetin complexes (FeQ) to the major allergen Alt a 1 from the fungus Alternaria alternata modifies the allergens immunogenicity. They found, that in in silico analyses Alt a 1 tetramers strongly bound to FeQ2 with theoretical Kd values reaching pico-molar levels. ApoAlt a 1 (without bound ligand) showed enzymatic activity as trimer or tetramer, which was not observed for holoAlt a 1 (with bound ligand). In human PBMCs holoAlt a 1 but not apoAlt a 1 significantly decreased surface expression levels of CD86 and HLADR, which may interfere with presentation of allergen-derived peptides.

In summary, these results suggest that nutritional iron-complexes such as FeQ2 alter the function of Alt a 1, resulting in reduced human immune responses. These results are interesting, since they support the theory, that Alt a 1 only becomes immunogenic (and probably allergenic) under nutritional deprivation of the fungus.

The manuscript contains findings interesting for the readers of IJMS, but some points have to be addressed before the manuscript is of sufficient quality for publication

Major comments:

Fig. 1: The shown KD values are derived from in silico docking analyses, correct? I think in this case, it needs to be pointed out in the figure, that these KD values are theoretical ones and not experimentally determined.

Have the authors further characterized their holo vs. apo proteins (purity in SDS-PAGE and native PAGE, apparent differences in MW upon addition of the ligand, level of endotoxin contamination (see also comment below), contamination with glucans derived from the expression system)?

Fig. 3d: Are calcein levels in the HoloAlt a 1 group significantly different from the medium group?

Fig. 3 f to h: Have the authors also determined uptake levels at e.g. 4°C to exclude unspecific binding of the fluorescently labelled proteins to the cell surface?

Figure 3: The authors should also determine cytokine secretion from THP-1 cells to see, if the observed differences in cytokine secretion (Fig. 5) are more pronounced in isolated monocytes

Fig. 5 With the exception of IL-5 and IL-13 the differences in cytokine secretion are rather low. The authors should think about including T cell analyses to check for the functional consequences in terms of T cell activation resulting from these cytokines.

Fig. 5 e to g: How can cytokine levels reach negative values for IL-4, IL-5, and IL-13?

Line 356: Endotoxins have strong effects on the used THP-1 cells. In the methods part of the paper, the endotoxin content of the used Alt a 1 is indicated as 2.07 EU/ml. However, without knowing the concentration of the protein, this information has little value. The authors should give the endotoxin content as EU/ mg protein and include controls containing the respective amounts of endotoxin in buffer to exclude LPS effects in their experimental system.

Have the authors determined whether holo- and apoALt a 1 may differ in their effect on THP-1 cells and PBMCs in terms of cytotoxicity? Increased cytotoxicity might also explain for the reduced expression of the investigated surface markers in Fig. 4.

Minor comments:

Figure 1a-d: I was unable to identify the “iron shown as a grey sphere”. Would it be possible to choose a color with more contrast?

Figure 1e: Is it possible to improve the resolution of the table in Figure 1?

Line 146: Can the potential pathogenicity just be inferred from protease activity? Or are there additional factors that contribute to Alternaria pathogenicity?

The manuscript would benefit from language editing by a native speaker. Some examples:

Line 28: add space between “activity” and “against”

Line 47: remove space after “radiation”

Line 51: remove space after “9]”

Line 56: add space between “11” and “beta”

Line 57: remove one of the two commas at the end of the line

Line 63: add comma after ”21]”

Lines 70 to 72: Check sentence, it seems incomplete. What can the allergen exert?

Line 76: I would suggest to replace “outstanding” with a more scientific term such as “very high affinity”

Line 83: remove space after “iron-”

Line 137: replace “addition” with “additional”

Line 138: “oligomerization or recombinant Alt a 1” should probably mean “oligomerization of recombinant Alt a 1”

Line 232: replace “was” with “were”

Line 261: replace “respected” with “respective”

Line 263: suggest to replace “outstanding” with “nanomolar”

Line 266: replace “antibody” with “antibodies”

Line 272: add comma after “further”

Line 294: replace “skills” with “capacity”

Line 403 “Am” should probably mean “an”

Line 444: replace “,” before “For” with “.”

Paragraph 4.11. Statistical analysis: Please check and grammatically revise the whole abstract.

Author Response

Review report: Nutritional provision of iron complexes by the major allergen Alt a 1 to human immune cells decreases its presentation

The authors investigated whether binding of iron quercetin complexes (FeQ) to the major allergen Alt a 1 from the fungus Alternaria alternata modifies the allergens immunogenicity. They found, that in in silico analyses Alt a 1 tetramers strongly bound to FeQ2 with theoretical Kd values reaching pico-molar levels. ApoAlt a 1 (without bound ligand) showed enzymatic activity as trimer or tetramer, which was not observed for holoAlt a 1 (with bound ligand). In human PBMCs holoAlt a 1 but not apoAlt a 1 significantly decreased surface expression levels of CD86 and HLADR, which may interfere with presentation of allergen-derived peptides.

In summary, these results suggest that nutritional iron-complexes such as FeQ2 alter the function of Alt a 1, resulting in reduced human immune responses. These results are interesting, since they support the theory, that Alt a 1 only becomes immunogenic (and probably allergenic) under nutritional deprivation of the fungus.

The manuscript contains findings interesting for the readers of IJMS, but some points have to be addressed before the manuscript is of sufficient quality for publication

 We thank the reviewer for the benevolent comments.

Major comments:

Fig. 1: The shown KD values are derived from in silico docking analyses, correct? I think in this case, it needs to be pointed out in the figure, that these KD values are theoretical ones and not experimentally determined.

Indeed, the findings are to in silico calculations. We amended the figure legend for better accuracy including that the calculated affinities derived from in silico docking of quercetin in conjunction with iron to mon-, di-, tetra- and hexameric Alt a 1.

Have the authors further characterized their holo vs. apo proteins (purity in SDS-PAGE and native PAGE, apparent differences in MW upon addition of the ligand, level of endotoxin contamination (see also comment below), contamination with glucans derived from the expression system)?

Based on your excellent suggestions, we performed silverstainings with cooked and uncooked samples of apo- and holoAlt a 1 and in the presence of Zn and Ca. Indeed, when using uncooked samples as in the zymography experiments, Alt a1 is present rather in a oligomeric state that might explain the results obtain by the Casein zymography, whereas we could not observe real significant size differences neither upon addition of iron-quercetin complexes (holoAlt a1) nor in in the presence or absence of Zn/Ca. We now included in the revised Figure 3a the respective silverstainings.

In regard to endotoxin-contamination apo- and holoAlta1 were throughout all experiments from the same batch. To generate apo/holoAlt a1, recombinant protein of the same batch was used, dialysed against the iron-chelator deferoxamine (to remove potential already-bound ligands) and generate apoAlt a1.  Subsequently ligands were added to apoAlt a1 to generate holoAlt a 1.

We though used two different batches of recombinant Alt a 1. All cellular experiments were conducted with a steril-filtered batch containing 2.07 EU/ml endotoxin reference E.coli O55:B5 in 1.1mg/ml, thus containing 1.89 EU/mg protein. Proteolytic analysis was conducted with a second batch (conc. 3.1mg/ml, endotoxin-levels 54.9 EU/ml = 17.7 EU/mg protein). We added in the material and method section a paragraph, detailing the different batches used and the respective endotoxin-levels.

Batches were sterile filtered (0.22µM) before use. In this manuscript two different batches of recombinant Alt a 1 were used. PBMC analysis were conducted with Alt a 1 batch 1 (conc. of stock: 1.1mg/ml containing 2.07 EU/ml endotoxin), and zymography was conducted with batch 3 (conc. of stock: 3.1mg/ml, 54.9 EU/ml).

We also added additional lines in the method section for the silverstaining analysis as followed:

10µl recombinant apo- and holoAlt a 1 in 10mM Hepes-buffer/150mM NaCl pH 6.5 (1mg/ml) with or without 2µl  10x Zn/Ca Tris Buffer ( 500mM Tris, 10mM ZnCl2, 50mM Ca) were mixed with 5µl  non-reducing 4x sample buffer pH 6.8 . Samples were left uncooked or were cooked for 5 min at 95°C before they were separated on Mini-Protean TGXStain-Free Gel 4-20% (Bio-Rad 4568095) electrophoresis. Gels were subsequently fixed and silverstained upon addition of thiosulfate solution (0.02%), silver nitrate reagent (0.02% silvernitrate, 0.02% formaldehyde) , developing in 3% sodium carbonate containing 5mg/l thiosulfate and stopping development with 0.5% glycine solution.”

Fig. 3d: Are calcein levels in the HoloAlt a 1 group significantly different from the medium group?

Indeed, it is still not significant. We matched data and used post-hoc Holm-Sidak’s multiple comparison test. Here, medium vs holoAlt a1 has a p-Value of 0.0600.

Fig. 3 f to h: Have the authors also determined uptake levels at e.g. 4°C to exclude unspecific binding of the fluorescently labelled proteins to the cell surface?

Thanks for the very constructive comment and valid point. In the previous manuscript version, we did not perform binding at 4°C. However, we repeated in the meanwhile these experiments and show attached medium/apo-/holo-Alta1 binding at 4°C versus 37°C, we included the 4°C values in the new figure 3 f showing % binding and 3g showing the mean intensity of Thp1-cells incubated with APC-labelled Alt a 1.

Figure 3: The authors should also determine cytokine secretion from THP-1 cells to see, if the observed differences in cytokine secretion (Fig. 5) are more pronounced in isolated monocytes.

Based on your suggestions, we repeated stimulation with apo/holoAlta1 in Thp1-cells and measured IL6, IFNg,  IL10 and IL13. However, the values obtained were still very, very low upon overnight incubation of  Thp1 cells in FCS-free medium and thus only attached for the reviewer’s  information. We thus may want to keep Figure 5 only depicting cytokines of PBMCs stimulations.

Fig. 5 With the exception of IL-5 and IL-13 the differences in cytokine secretion are rather low. The authors should think about including T cell analyses to check for the functional consequences in terms of T cell activation resulting from these cytokines.

Indeed, the levels are very low, however one must consider that cells were stimulated only for 18h with medium containing hardly any nutrients (as no FCS was in the culture medium). We do have technical limitations when studying the impact of iron with proteins/allergens as – though iron is essential for cell survival- it has to be omitted in the cell culture medium to not falsify the data. After 18h, the major cytokine source are predominantly monocytes and/or B cells and most cells are still living. However usually cytokine synthesis for T cells require longer time periods (72h or 96h) and thus cell death due to starving occur. As this falsify the data obtain for T cells, we are unsure how to detangle the impact of the protein versus starvation on the data analysis.

Fig. 5 e to g: How can cytokine levels reach negative values for IL-4, IL-5, and IL-13?

We now “forced” the standard curve to go to the lowest obtain values and share now the corrected values for IL4, IL5 and IL13 and amended the Figure 5 accordingly.

Line 356: Endotoxins have strong effects on the used THP-1 cells. In the methods part of the paper, the endotoxin content of the used Alt a 1 is indicated as 2.07 EU/ml. However, without knowing the concentration of the protein, this information has little value. The authors should give the endotoxin content as EU/ mg protein and include controls containing the respective amounts of endotoxin in buffer to exclude LPS effects in their experimental system.

Thanks again for the valid comment. We added the characteristics of the batches in the method section in greater detail. The EU of Alt a 1 was with 2.07 EU/mg protein much lower than previously measured endotoxin-levels of commercially available beta-lactoglobulin having 3000 EU/mg protein, alpha-lactalbumin with approx. 500 EU/mg protein and even with endotoxin-low caseins, which rendered in our hand 4 EU/mg protein.

The PBMCs were stimulated with 2.5µM Alt a 1 (75µg/ml) corresponding to 0.15 EU/ml, which we consider very low. Moreover apo-/holoAlt a1 were from the same batch. Based on our experience also with other proteins, we are thus confident that the differences between apo- and holoAlt a 1 cannot be explained by the impact of endotoxins.

Have the authors determined whether holo- and apoALt a 1 may differ in their effect on THP-1 cells and PBMCs in terms of cytotoxicity? Increased cytotoxicity might also explain for the reduced expression of the investigated surface markers in Fig. 4.

Thanks again for the constructive remarks. Indeed, in our gating strategy we only assessed “living cells”. However, we now add data to show the impact of 18h incubation under starving conditions of medium/apoAlta1 and holoAlta1 on cell death, which we include now as Supplementary Figure 1 in the submitted manuscript and add them also here for the reviewer’s information. In brief, we were not able to discern increased cell death in the population upon either incubation of apo- or holoAlta1.

Minor comments:

Figure 1a-d: I was unable to identify the “iron shown as a grey sphere”. Would it be possible to choose a color with more contrast

Actually, the caption was misleading, as iron was shown as an orange/brown sphere. We thank the reviewer to point this out!

Figure 1e: Is it possible to improve the resolution of the table in Figure 1?

We resized and enlarged the images in Figure 1 for better visibility.

Line 146: Can the potential pathogenicity just be inferred from protease activity?

As stated in the introduction Alternaria is an opportunistic pathogen. However, though more studies have to be conducted to know with certainty, the protease activity might be one factor in the increased potential pathogenicity of Alternaria.

Or are there additional factors that contribute to Alternaria pathogenicity?

Beside showing a protease activity, they are also able to secrete toxins. There are certainly, multiple triggers and mechanisms, that is able to turn Alternaria into a pathogen. More studies are needed to elucidate the additional factors. 

Comments on the Quality of English Language

The manuscript would benefit from language editing by a native speaker. Some examples:

 Line 28: add space between “activity” and “against”

Line 47: remove space after “radiation”

Line 51: remove space after “9]”

Line 56: add space between “11” and “beta”

Line 57: remove one of the two commas at the end of the line

Line 63: add comma after ”21]”

Lines 70 to 72: Check sentence, it seems incomplete. What can the allergen exert?

Line 76: I would suggest to replace “outstanding” with a more scientific term such as “very high affinity”

Line 83: remove space after “iron-”

Line 137: replace “addition” with “additional”

Line 138: “oligomerization or recombinant Alt a 1” should probably mean “oligomerization of recombinant Alt a 1”

Line 232: replace “was” with “were”

Line 261: replace “respected” with “respective”

Line 263: suggest to replace “outstanding” with “nanomolar”

Line 266: replace “antibody” with “antibodies”

Line 272: add comma after “further”

Line 294: replace “skills” with “capacity”

Line 403 “Am” should probably mean “an”

Line 444: replace “,” before “For” with “.”

Paragraph 4.11. Statistical analysis: Please check and grammatically revise the whole abstract.

We revised the manuscript and hope that all reviewer’s comments could be met.

Reviewer 2 Report

The authors use recombinant Alt a 1 produced in P pastoris to examine the proteins’ structure and function.  They use several different assays, but the data in many cases lack the impact needed to support their conclusions.  There are several instances where the text is a bit difficult to interpret and could be improved. 

In the introduction some discussion as quercetin as an iron chelating molecule would be useful. 

Figure 1 represents in silico prediction, and the legend should indicate that these are ‘predicted’ interactions and values in the table (e).  The images and text in c and d should be enlarged, they are very difficult to interpret. 

There are two intramolecular cysteine disulfide bonds within each monomer in the 3VOR pdb file, please explain/clarify the sentence on lines 56-58 indicating only ‘a single disulfide bond’.  It’s not clear from that structure where the intermolecular disulfide bond forms in the dimer.  Is this assertion only from SDS-PAGE or from a previous/additional crystal structure? 

Add “(FeQ2)” following the words “ iron quercetin complex” on line 104 

Labeling and identification of the peaks in Figure 2a must be improved.  What is the maximum at 460 nM representing?  It’s not clear from the text, is the 460 nm maximum for quercetin?  Can the authors explain the long shoulder on the left side of the graph for FeQ2+Alt a 1 data lines?  As a control, the authors should include an Fe alone spectra in the same graph.  There are no error bars on the graph and it is hard to agree with the statement that binding is ‘proven’ on line 106.  The difference does not appear to be clearly concentration dependent, but there is an effect at the 8 µM Alt a 1 concentration at 460 nM in peak intensity.  Binding to quercetin alone is not as convincing, but is similar to the effect seen with the FeQ2 + 2 µM Alt a 1 and FeQ2 + 4 µM Alt a 1. 

Figure 2D shows no difference in size between the holo and apo Alt a 1 molecules, but the authors state on 118—119 ‘a greater degree towards oligomerization was apparent (Figure 2d).’  This data does not  appear to support the conclusion they make.  Is there a corresponding increase in a peak of larger than ~2nM size?  Please explain the y-axis, is it % mass from a total of 100%?  If so, where is the balance of the signal? 

Figure 3 is confusing/hard to interpret.  Is the recombinant Alt a 1 dimer or monomer
(of 12.5 kDa) migrating at 25 kDa?  What is the expected size of the recombinant Alt a 1 monomer?  Why are such low concentrations/amounts of recombinant protein used in Figure 3a?  Is the protein pure?  Can the authors provide some assessment of their recombinant Alt a 1 purity?  Is there mass-spec data to indicate if there are contaminating proteins that may represent the 15 other slower migrating bands in the figure 3a image?  These bands could be contamination that is responsible for the degradation of casein and/or gelatine observed in Figure 3b and c?  Alt a 1 has an (AA1)-like fungal toxin domain, but does Alt a 1 have any sort of peptidase domain, or other motif that would suggest it cleaves/metabolizes proteins/peptide bonds within casein, gelatine, or other protein substrate?    

The zymography figure is interesting, but Figure 3 b and c are difficult to visualize and the authors should consider inverting them to a white background and black/grey signal, they are not convincing as presented and inverting the signal would make the corresponding quantification/graphs of the data easier to interpret.  While the comparison between apo and holo Alt a 1 provide intriguing results, it is difficult to interpret.  Do figure 3 b and c only show activity from the Alt a 1 protein?  It’s very difficult to understand how the trimer (~37.5 kDa gelatine degradation versus ~110 kDa casein) or Alt a 1 hexamer have different activities/substrate specificities and that the substrate may induce/cause them to form different oligmomers with different peptidase activities.  Can the authors try to explain this?  Where is the no Alt a 1 (apo or holo) control?  The difference in figure 3d appears to be very slight and is not ‘significantly decreased’ as the authors state.  Can the authors demonstrate that Alt a 1 was taken up by the cells?   Figure 3e does not appear to be a linear response, yet the authors conclude as such.  Can they explain why?  They y-axis of figure 3 f, g, and h would suggest the observed effects are very slight.  There is no trend in figure 3 f, there is only 1 point of data presented to consider. 

The graphs in figure 4 c, d, and e are not easily interpreted and don’t appear to support the conclusions (some increase and some decrease) the authors state. 

The data in figure 5 does not support a ‘significant’ change for IL6 (nor for TNFα or IL10) as stated by the authors on lines 228-234.  There may be some slight differences in IL4, IL5, and IL13 compared to the control, but the data presented do not appear to support/argue for significant differences between apo and holo Alt a 1.   

The writing should be improved prior to publication to ease reader comprehension. 

There are several instances where the text is a bit difficult to interpret and could be improved. 

Author Response

Comments and Suggestions for Authors

The authors use recombinant Alt a 1 produced in P pastoris to examine the proteins’ structure and function.  They use several different assays, but the data in many cases lack the impact needed to support their conclusions.  There are several instances where the text is a bit difficult to interpret and could be improved. 

In the introduction some discussion as quercetin as an iron chelating molecule would be useful. 

We thank the reviewer for the suggestion and we included now following sentences for quercetin to support its function as iron chelator in line 64-71

Indeed, quercetin can serve as an iron source for commensal and opportunistic microbial pathogens [22] and its anti-oxidative  and ROS-scavenging properties is reported to be greater in complex with iron than without iron[23-25]. The reported complex stability constant (log ß) for quercetin is 44.2 at physiological pH[26]; and thus comparable to the iron affinity of the strongest known microbial catechol-siderophore enterobactin having a complex stability constant of 49 at physiological pH[27]. Also, in humans its impact in human iron homeostasis is well recognized [25, 28, 29] with therapeutic strategies usually exploiting [30] its function as iron-chelator [29, 31-33]”

Figure 1 represents in silico prediction, and the legend should indicate that these are ‘predicted’ interactions and values in the table (e).  The images and text in c and d should be enlarged, they are very difficult to interpret. 

We agree with the reviewer’s comment. We now emphasize in the Figure legend that “… the calculated affinities derived from in silico docking of quercetin in conjunction with iron to mon-, di-, tetra- and hexameric Alt a 1”. Moreover, we enlarged the images in Figure 1 for better visibility.

There are two intramolecular cysteine disulfide bonds within each monomer in the 3VOR pdb file, please explain/clarify the sentence on lines 56-58 indicating only ‘a single disulfide bond’.  It’s not clear from that structure where the intermolecular disulfide bond forms in the dimer.  Is this assertion only from SDS-PAGE or from a previous/additional crystal structure? 

The 'single disulfide bond' mentioned in this sentence was indeed intended to refer to the intermolecular one, which is described in more detail in reference [13]. The findings there are based on the crystal structure (3v0r) and SDS-PAGE; see e.g. Figures 1 and 2 in [13].

Add “(FeQ2)” following the words “ iron quercetin complex” on line 104 

Labeling and identification of the peaks in Figure 2a must be improved.  What is the maximum at 460 nM representing?  It’s not clear from the text, is the 460 nm maximum for quercetin?  Can the authors explain the long shoulder on the left side of the graph for FeQ2+Alt a 1 data lines?  As a control, the authors should include an Fe alone spectra in the same graph.  There are no error bars on the graph and it is hard to agree with the statement that binding is ‘proven’ on line 106.  The difference does not appear to be clearly concentration dependent, but there is an effect at the 8 µM Alt a 1 concentration at 460 nM in peak intensity.  Binding to quercetin alone is not as convincing, but is similar to the effect seen with the FeQ2 + 2 µM Alt a 1 and FeQ2 + 4 µM Alt a 1. 

As we do have a long-standing experience with quercetin and iron and regularly perform these analyses, we apologize for not detailing enough the UV-VIS spectral analysis. Indeed, quercetin has per se a yellow color, which result at physiological pH of a single maximum at about 340nm. In contrast, when in complex with iron at a pH > 6.5, the color turns brown and due to the interaction of the catechol-group with the d-orbitals of ferric iron a second maxima usually about 450nm (exact maxima dependent on the salinity, buffer system and pH) appears [1-3]. Particularly the “second, iron peak” is affected upon adding apoAlt a1 as - though a surplus of this complex is present (40µM FeQ2 – light blue line) - upon addition of 2, 4 and 8 µM Alt a 1 (lines in darker blue hues) a significant concentration-dependent diminishment of this iron peak occur, which again is visibly by eye only and leads to a discoloration.

As this “discoloration” is routinely assessed by us– indeed to prove binding- and is also done by other groups to show the iron-scavenging proteins such as LCN2/NGAL[4] and novel heme-binding proteins[5], we indeed consider the assay valid and to prove binding. However, we amended the figure added iron alone as control (black line) and explained in greater detail Figure 5a as followed in line 114-119 :

Indeed, quercetin has per se a light yellow colour, which result in  a spectrum at physiological pH peaking with a single maximum at about 340nm. In contrast, when in complex with iron (at a pH > 6.5), the colour turns brownish and due to the interaction of the catechol-group with the d-orbitals of ferric iron a second maxima usually about 450nm appears [1-3]. Binding to FeQ2 could be proven, because this 2nd absorption maximum diminished concentration-dependently upon addition of Alt a 1 (empty protein, blue lines) (Figure 2 a).”

Figure 2D shows no difference in size between the holo and apo Alt a 1 molecules, but the authors state on 118—119 ‘a greater degree towards oligomerization was apparent (Figure 2d).’  This data does not  appear to support the conclusion they make.  Is there a corresponding increase in a peak of larger than ~2nM size?  Please explain the y-axis, is it % mass from a total of 100%?  If so, where is the balance of the signal? 

Indeed, the DLS-data are not very strong as it only can give indications on oligomerization and do not show distinct peaks. The y-axis represent % intensity that after Stetefeld et al. [6] can be used to study the homogeneity of proteins as well as protein-protein interaction. Here %intensity depicts approximately the size6 distribution according to the Rayleigh approximation. As such, we changed in the new revised Figure 2d the log-scale to linear scale to better visualize that upon FeQ2 addition the molecular size shifts towards greater size (with 2nm approximately being the monomer and 2.8nm representing a dimer). We interpret thus that upon FeQ2 addition, Alt a 1 tends to form more oligomers.  We revised in the manuscript  and added following lines 129-133:

“. Here, %intensity was used  as an approximation for the size6 distribution in the solution [6] . With and without a ligand, the quaternary state was similar between pH 5.5 and 7.2 in apo- and holoAlt a 1 (Figure 1c), however when comparing % intensity of  apoAlt a 1 and holoAlt a 1 at pH 6.5 or 7.2  on  a linear scale, a greater degree towards oligomerization was apparent”

Figure 3 is confusing/hard to interpret.  Is the recombinant Alt a 1 dimer or monomer
(of 12.5 kDa) migrating at 25 kDa?  What is the expected size of the recombinant Alt a 1 monomer?  

The size of monomeric Alt a 1 is 15242.88, with the dimer running at 25 kDa.

Why are such low concentrations/amounts of recombinant protein used in Figure 3a? 

We revised Figure 3 and added additional silverstainings of recombinant Alt a 1 (in higher concentrations) with or without heat-treatment and separated under non-reducing conditions in the presence of absence of FeQ2 and/or Zn/Ca to gain further insight on the quarternary state of   Alt a 1.

 Is the protein pure?  Can the authors provide some assessment of their recombinant Alt a 1 purity?  Is there mass-spec data to indicate if there are contaminating proteins that may represent the 15 other slower migrating bands in the figure 3a image? 

Indeed, the recombinant protein from was transfected cell supernatant were purified with anion exchange chromatography using 10 mM Na-phosphate buffer (pH 7.5). SDS-PAGE and circular dichroism (CD) spectroscopy were used to verify protein purity, identity and secondary structure. Measurement of endotoxin content was done by EndoZyme recombinant Factor C Endotoxin Detection Assay (Biomerieux, France) and total protein content by BCA assay. However, no mass spectrometric analysis was performed, but functional test with patients’ IgE was performed to assess functionality of Alt a 1. Batches were sterile filtered (0.22µM) before use. In this manuscript two different batches of recombinant Alt a 1 were used. PBMC analysis were conducted with Alt a 1 batch 1 (conc. of stock: 1.1mg/ml containing 2.07 EU/ml endotoxin), and zymography was conducted with batch 3 (conc. of stock: 3.1mg/ml, 54.9 EU/ml)

 These bands could be contamination that is responsible for the degradation of casein and/or gelatine observed in Figure 3b and c? 

Indeed, this is unlikely as then also in the gelatine or casein zymographie degradation at this low molecular weight should be visible. All experiments were at least repeated twice, but usually three times ensuring reproducibility of the results.

 Alt a 1 has an (AA1)-like fungal toxin domain, but does Alt a 1 have any sort of peptidase domain, or other motif that would suggest it cleaves/metabolizes proteins/peptide bonds within casein, gelatine, or other protein substrate?    

Upon your valid question, we searched Interpro (https://www.ebi.ac.uk/interpro/search/text/metalloproteases/?page=1#table), which listed 191 entries for metalloproteases, but only described very few proteolytic domains/motifs.  One conserved motif is HExxH has been shown in crystallographic studies to form part of the metal-binding site, however it is not present in our protein sequence. As such, we can only speculate on the specific enzymatic domain. We though have to emphasize that its enzymatic function has also been described by at least two other work groups [7, 8]

The zymography figure is interesting, but Figure 3 b and c are difficult to visualize and the authors should consider inverting them to a white background and black/grey signal, they are not convincing as presented and inverting the signal would make the corresponding quantification/graphs of the data easier to interpret.  While the comparison between apo and holo Alt a 1 provide intriguing results, it is difficult to interpret. 

We thank for the benevolent comment, we now inverted the images to improve visibility and hope that thereby we could meet the reviewer’s critic. 

Do figure 3 b and c only show activity from the Alt a 1 protein?  It’s very difficult to understand how the trimer (~37.5 kDa gelatine degradation versus ~110 kDa casein) or Alt a 1 hexamer have different activities/substrate specificities and that the substrate may induce/cause them to form different oligmomers with different peptidase activities.  Can the authors try to explain this?  

Indeed, we were also puzzled by these results. Based on the new performed silverstains, our data indicate that the hexameric/tetrameric form can function as peptidase, whereas the addition of Calcium and Zinc enables a trimeric form to exert proteolytic function. Indeed, we were only able to see any enzymatic function, when iron-quercetin was not added to the protein, despite that in the electrophoretic analyses they ran very similarly. However, also others have reported that the enzymatic function depended on the quarternary state [9][10] and some proteins such as human RecQ having dual enzymatic function depending on its quarternary state [11]. As such, redundant mechanism seems to exist, with our data indicating that Alt a 1 can exert different function dependent of the environmental context.  

Where is the no Alt a 1 (apo or holo) control?  The difference in figure 3d appears to be very slight and is not ‘significantly decreased’ as the authors state. 

Indeed, medium is the no Alt a 1 control in Figure 3d. Data were matched per experiments and statistically analysis with a p-Value below 0.05 considered significant.  

 Can the authors demonstrate that Alt a 1 was taken up by the cells?   Figure 3e does not appear to be a linear response, yet the authors conclude as such. 

The reviewer is correct, we did not see a linear response of quercetin-dependent AhR-activation, but upon addition of Alt a 1 always an increase in AhR activation.

 Can they explain why?  

In contrast to other allergens such as Bet v 1 or Bos d 5/BLG, Alt a 1 “behaves” differently as it does not seem to contribute its iron to the labile iron pool once in the cells (which is in stark contrast to Bet v1 or BLG, in which we see an increase of cytosolic iron). Similarly, also with quercetin - though Alt a 1 seems to transport quercetin into the cell, quercetin is not as  capable to induce AhR-activation. We hypothesize that one reason may be that Alt a 1 binds too strongly to iron quercetin complexes and thus not share its nutrient with the host cell.

They y-axis of figure 3 f, g, and h would suggest the observed effects are very slight.  There is no trend in figure 3 f, there is only 1 point of data presented to consider. 

We performed additional experiments at 4 and 38°C to proof that binding was not only on the plasma membrane, but that the fluorescent-labelled Alt a 1 was indeed within the cell.  We further want to point out that preincubation with the endocytosis inhibitors indeed could reduce Alt a 1 uptake. We do not claim that this completely abrogated uptake, but that uptake occurred partly via these channels. As the results were several times repeated by independent experiments we are confident of the obtained results.

The graphs in figure 4 c, d, and e are not easily interpreted and don’t appear to support the conclusions (some increase and some decrease) the authors state. 

We want to emphasize that in Figure 4, data of primary immune cells isolated from Alternaria allergic patients (with different gender and sensitization status) are depicted, which cannot compare with normal cell culture studies. As such, this data are very convincing and completely support the notion, that the ligand-status of this allergen deeply affects the maturation state of monocytic cells, and thus directly affect immunity.

The data in figure 5 does not support a ‘significant’ change for IL6 (nor for TNFα or IL10) as stated by the authors on lines 228-234.  There may be some slight differences in IL4, IL5, and IL13 compared to the control, but the data presented do not appear to support/argue for significant differences between apo and holo Alt a 1.   

The reviewer is correct, that the cytokine secretion is very modest, which we also state in the manuscript. However, this is due to the technical limitations that we face when investigating iron-mediated processes.  

We want to emphasize that in Figure 4, data of primary immune cells isolated from Alternaria allergic patients (with different gender and sensitization status) are depicted, which cannot

The writing should be improved prior to publication to ease reader comprehension. 

Comments on the Quality of English Language

There are several instances where the text is a bit difficult to interpret and could be improved. 

References

  1. Raza A, Xu X, Xia L, Xia C, Tang J, Ouyang Z. Quercetin-Iron Complex: Synthesis, Characterization, Antioxidant, DNA Binding, DNA Cleavage, and Antibacterial Activity Studies. J Fluoresc. 2016;26(6):2023-31.
  2. Roth-Walter F, Afify SM, Pacios LF, Blokhuis BR, Redegeld F, Regner A, et al. Cow's milk protein beta-lactoglobulin confers resilience against allergy by targeting complexed iron into immune cells. J Allergy Clin Immunol. 2021;147(1):321-34 e4.
  3. Regner A, Szepannek N, Wiederstein M, Fakhimahmadi A, Paciosis LF, Blokhuis BR, et al. Binding to Iron Quercetin Complexes Increases the Antioxidant Capacity of the Major Birch Pollen Allergen Bet v 1 and Reduces Its Allergenicity. Antioxidants (Basel). 2022;12(1).
  4. Bao G, Clifton M, Hoette TM, Mori K, Deng SX, Qiu A, et al. Iron traffics in circulation bound to a siderocalin (Ngal)-catechol complex. Nat Chem Biol. 2010;6(8):602-9.
  5. Khan D, Lee D, Gulten G, Aggarwal A, Wofford J, Krieger I, et al. A Sec14-like phosphatidylinositol transfer protein paralog defines a novel class of heme-binding proteins. Elife. 2020;9.
  6. Stetefeld J, McKenna SA, Patel TR. Dynamic light scattering: a practical guide and applications in biomedical sciences. Biophys Rev. 2016;8(4):409-27.
  7. Saenz-de-Santamaria M, Guisantes JA, Martinez J. Enzymatic activities of Alternaria alternata allergenic extracts and its major allergen (Alt a 1). Mycoses. 2006;49(4):288-92.
  8. Barnes CS, Pacheco F, Landuyt J, Rosenthal D, Hu F, Portnoy J. Production of a recombinant protein from Alternaria containing the reported N-terminal of the Alt a1 allergen. Adv Exp Med Biol. 1996;409:197-203.
  9. Segura-Pena D, Lichter J, Trani M, Konrad M, Lavie A, Lutz S. Quaternary structure change as a mechanism for the regulation of thymidine kinase 1-like enzymes. Structure. 2007;15(12):1555-66.
  10. Szigetvari PD, Muruganandam G, Kallio JP, Hallin EI, Fossbakk A, Loris R, et al. The quaternary structure of human tyrosine hydroxylase: effects of dystonia-associated missense variants on oligomeric state and enzyme activity. J Neurochem. 2019;148(2):291-306.
  11. Muzzolini L, Beuron F, Patwardhan A, Popuri V, Cui S, Niccolini B, et al. Different quaternary structures of human RECQ1 are associated with its dual enzymatic activity. PLoS Biol. 2007;5(2):e20.

Round 2

Reviewer 1 Report

the authors have addressed most of my comments.

the purity of the used Alt a 1 is rather low, the authors must comment on the other bands and how these might influence the obtained results.

the discussion is still overstating some of their results. E.g. the theortically computed affinity of the protein to the ligand is stated in a way that suggests that is was detemined experimentally (line 289).  The authors should indicate the theoretical anture of these results.

the manuscript would still benfit from further improvement. I have underlined some examples in the attached file

Author Response

the authors have addressed most of my comments.

the purity of the used Alt a 1 is rather low, the authors must comment on the other bands and how these might influence the obtained results.

We thank for the critical review;  and tried now to address the purity of Alt a 1 in the discussion section, line 302-306 as followed:

We do not attribute - but cannot completely exclude- that the displayed enzymatic ac-tivities of the recombinant Alt a 1, is due to remaining impurities in the same molecu-lar size and with similar physico-chemical properties as Alt a 1, because also other groups have reported that Alt a 1 displays enzymatic phosphatase, phosphoamidase and esterase -activity[78].

the discussion is still overstating some of their results. E.g. the theortically computed affinity of the protein to the ligand is stated in a way that suggests that is was detemined experimentally (line 289).  The authors should indicate the theoretical anture of these results.

Also here we amended the discussion section by emphasizing the “theoretical” measurements and including following lines 290-294:

Although the affinities have only been calculated by in silico means and just give an approximation on the “real” affinity, the calculated software Autodock Vina has been validated in numerous studies. Consequently, it is used nowadays as a tool for screen-ing potential protein ligand interactions and predicting their affinities [70-76].

We hope thereby that we could address all points raised.

Thanks again for the review and the excellent suggestions.
